# Validation of Multiaxial Fatigue Strength Criteria on Specimens from Structural Steel in the High-Cycle Fatigue Region

**DOI:** 10.3390/ma14010116

**Published:** 2020-12-29

**Authors:** František Fojtík, Jan Papuga, Martin Fusek, Radim Halama

**Affiliations:** 1Faculty of Mechanical Engineering, Department of Applied Mechanics, VŠB—Technical University of Ostrava, 17. listopadu 2172/15, 70800 Ostrava, Czech Republic; martin.fusek@vsb.cz (M.F.); radim.halama@vsb.cz (R.H.); 2Department of Mechanics, Biomechanics and Mechatronics, Faculty of Mechanical Engineering, Czech Technical University in Prague, Technická 4, 16607 Prague, Czech Republic; jan.papuga@fs.cvut.cz; 3Center of Advanced Aerospace Technology, Faculty of Mechanical Engineering, Department of Instrumentation and Control Engineering, Czech Technical University in Prague, Technická 4, 16607 Prague, Czech Republic

**Keywords:** multiaxial fatigue, high-cycle fatigue, multiaxial fatigue experiments, S-N curve approximation

## Abstract

The paper describes results of fatigue strength estimates by selected multiaxial fatigue strength criteria in the region of high-cycle fatigue, and compares them with own experimental results obtained on hollow specimens made from ČSN 41 1523 structural steel. The specimens were loaded by various combinations of load channels comprising push–pull, torsion, bending and inner and outer pressures. The prediction methods were validated on fatigue strengths at seven different numbers of cycles spanning from 100,000 to 10,000,000 cycles. No substantial deviation of results based on the selected lifetime was observed. The PCRN method and the QCP method provide best results compared with other assessed methods. The results of the MMP criterion that allows users to evaluate the multiaxial fatigue loading quickly are also of interest because the method provides results only slightly worse than the two best performing solutions.

## 1. Introduction

To validate the multiaxial fatigue strength criteria, the prediction results should be compared with experimental results. Any experimental campaign that would cover the mean normal stress effect, the mean shear stress effect, phase shift effect, etc., in various lifetimes for a single material (or more materials) presents a lengthy and costly process. The validation is thus often realized on experimental data retrieved from other sources: conference papers, papers in journals, books, PhD theses or technical reports. If such sources are used, they must be carefully verified, as to whether the data to be adopted are credible and usable for such validation.

The prediction quality of multiaxial fatigue strength criteria is often assessed on experiments with various load combinations of push–pull and torsion [1,2,3,4]. By some experiments, axial loading of specimens can be induced by bending, which causes non-constant stress distribution over the cross-section [5,6,7,8,9]. Relatively rarely, the inner pressure is applied on hollow specimens [10,11,12]. The stress gradient differs in this load mode—it is higher at the inner surface and lower at the outer surface. It is, therefore, important to know, which of those two surfaces is critical due to the other co-acting load channels, and both surfaces should be evaluated in some cases. The pressure is usually acting as a constant load. Such setup is simpler, because it allows the experimenter to run the desired load history quicker. If more load channels are superposed, the multiaxial stress state is likely to be induced. Such more complicated combinations are very interesting for validation purposes. Experimental sets comprising multiple various load cases on different specimen designs and sizes, that also provide a detailed information on material properties to tune the multiaxial fatigue strength criteria, are published only rarely. The experiments described in this paper cover various combinations of push–pull, bending, torsion, inner and outer pressures. The broad spectrum of various multiaxial load combinations with varying stress ratios on individual channels are an important condition for any adequate validation of calculation methods used by different multiaxial fatigue strength criteria.

One of the few papers, which looks for a general solution covering more different load modes in one campaign is the paper by Morel and Palin-Luc [13], in which they propose the use of the non-local model averaging the stress quantities over the critical volume. The paper by Papuga et al. [14] does not treat the same problem more generally and simply uses the axial load modes in the analysis according to assumed stress distribution. If multiaxial loading includes bending, the plane bending fatigue strengths are used in analyses. If it involves any other load case than bending, the fatigue strengths relevant to push–pull are used. This paper presents an extensive experimental campaign that was realized on hollow specimens made from ČSN 41 1523 structural steel. There are results of 24 uniaxial and multiaxial load cases of very diverse setups. The paper validates several multiaxial fatigue strength criteria on fatigue strengths obtained at 750,000 cycles from the Kohout-Věchet approximations [15] for each load case. The new MMP method usable for multiaxial fatigue strength analysis is published in [14] for the first time. As MMP is an extension of the Manson–McKnight criterion (MMK) [16,17], its biggest advantage is the simplicity of the computational analysis. It can be easily run using a common spreadsheet program such as MS Excel. The newly introduced MMP criterion significantly improves the prediction quality found for the MMK solution to a point that the MMP criterion could reach the quality of the output comparable with much more complex multiaxial fatigue strength criteria. Papuga et al. used the same data set in [18] to describe the validation results by selected multiaxial fatigue strength criteria at three additional lifetimes (the analyses were run between 100,000 and 750,000 cycles).

The current paper increases the scope of tested load cases from 24 to 34. This enlargement brings along some important load cases missing previously for some sizes of specimens—above all, the reversed torsion on smaller specimens or the repeated bending load case. These previously non-existent load cases had to be in some way substituted in the previous papers. Their inclusion into the computational scheme should result in a more consistent validation process.

The recent paper by Karolczuk et al. [19] opened a question, whether the same material parameters weighting the effect of stress parameters in the multiaxial criteria could be used over bigger ranges of lifetimes. They proved that the actual material parameters valid for the given final lifetime should result in a superior output than some fixed constant parameters could provide. To confirm or to deny that finding, the validation campaign on all 34 load cases is performed in this paper on fatigue strengths derived at seven different lifetimes spanning from 100,000 cycles to 10,000,000 cycles. To also cover the high lifetime levels, a special approximation FF formula is adopted in this paper. With all these changes involved, the paper focuses on validating 11 different multiaxial fatigue strength criteria of various types of solutions in order to assess their credibility for an accurate fatigue strength evaluation.

To reach this goal, the experimental campaign is first described in Section 2, including also the regression models used to derive the fatigue strengths from the S-N curves. The various multiaxial models processed in the validation are described in Section 3, together with the way the quality of the regression is assessed. Section 4 discusses the obtained results and Section 5 concludes the outcome of the presented paper.

## 2. Experiments and Processing of Their Results

### 2.1. Material and Specimens

The specimens were manufactured from ČSN 41 1523 structural steel (equivalent to S355JR or St52-3) delivered in bars retrieved from the single T31052 melt. The static material properties are provided in Table 1, and the chemical composition can be found in Table 2.

Three different specimen types: S1, S2 and S3 have already been used in [14]. The diameters (*D*—outer diameter, *d*—inner diameter) in the critical cross-sections were: (S1) *D* = 11 mm and *d* = 8 mm, (S2 and S3) *D* = 20 mm and *d* = 18 mm—see Figure 1. The newly introduced specimen type S4 is also shown in the same figure with its cross-sectional parameters slightly modifying the S1 configuration: *D* = 12 mm and *d* = 8 mm. All specimens were polished on the outer surface, while the inner surface was reamed.

### 2.2. Load Cases

The original experimental campaign from [14] is extended in this paper by 10 further new load cases. The summary of all load cases imposed in the total of 34 configurations is provided in Figure 2 and Figure 3. The detailed description of individual tests sets, of their types, of applied test frequencies and of geometric parameters of used specimens can be found in Table 3.

The new experimental load cases concern:Load case FF041—repeated plane bending.Load cases FF048-FF051—repeated bending with constant inner pressure imposed in the mode of a pressure vessel.Load cases FF062-FF063—fully reversed push–pull combined with inner pressure.Load cases FF064-FF065—repeated push–pull combined with inner pressure.Load case FF092—fully reversed torsion on S4 specimen (see Figure 1).

The S4 specimens were not manufactured on purpose after publishing [14]—they were prepared before the whole test campaign in the moment, when the search for an optimum geometry of specimens was targeted. As other specimen types were later selected for testing, the results of S4 specimens tested in fully reversed torsion were put aside. They were again uncovered, when the lack of the S-N curve for fully reversed torsion on smaller specimen type, S1, became obvious.

Figure 2 provides schematic drawings explaining the way the individual specimens were loaded for specific load cases marked FFXXX, where XXX is replaced by a unique ID number for each load case. All load cases were run under the load control. The ranges of forces, moments and pressures applied to individual load cases are provided in Table 4, and the information on the phase shift between the axial load channel and the torsion load channels accompanies them there.

To obtain the described load cases, various experimental machines had to be used as noted in Table 3, and different special fixtures to impose the desired load configurations had to be applied. Figure 3, left, depicts the example of the fixture model with the test specimen in grips as used for the load case of reversed bending marked as FF040. The combined bending and torsion loading as applied in FF044 test series is depicted in Figure 3, right. The FF046 load case combining the reversed bending with constant internal pressure can be found in Figure 4, left. The test specimens loaded by reversed torsion with inner and outer pressure in the FF059 test case can be seen in Figure 4, right. These tests were performed on the reconstructed and modernized biaxial testing machine Schenck type PWXN, which is originally equipped by the control of torque. It is extended by the possibility to apply additional axial constant force. Table 3 refers to this machine type by number 2. Number 1 in Table 3 corresponds to the biaxial servohydraulic pulsator INSTRON 8802. Another used testing machine is the biaxial servohydraulic pulsator LABCONTROL 100 kN/1000 Nm, which is equipped by a combined hydraulic actuator able to impose push–pull and torque. This machine was derived during the reconstruction of the original INOVA ZUZ 200 machine. It is marked by number 3 in Table 3. Number 4 in Table 3 concerns the uniaxial hydraulic pulsator INOVA FU-63-930-V1.

The necessary input into all multiaxial fatigue strength criteria are local stresses. The purely elastic material response is assumed to derive them. To locate the hot-spot on more complicated testing specimens for various superposed load channels’ acting, the finite element (FE) solution is necessary in order to deliver stress tensor components induced by individual load channels. All test cases and specimens were modeled and computed within the Ansys FE-solver. The stress tensors were obtained for unit loads acting on individual load channels, and the obtained stress components were then multiplied by the factor related to the ratio between the actual load and the unit load. Experiments FF062-FF065 are special, because the pressure causes higher (tangential) stress on the inner surface, while the stress response to axial loading induces more or less uniform axial stress distribution over the cross-section. For these specimens, thus, inner and outer surfaces were evaluated in the stress analysis and also in the subsequent fatigue analysis.

During the fatigue tests, responses to individual load channels were measured by certified sensors, which are regularly checked by the Czech Metrology Institute. To set up the load parameters by individual tests, strain gages were installed on chosen tested specimens; see the examples in Figure 5. The strain gages provided the information on strains and stresses attained on the surface of test specimens for individual load cases. These data items then could be compared with results of the finite element analyses to verify the applied boundary conditions.

### 2.3. Regression Analyses

In the study by [14], experimental results were processed to obtain the regression S-N curves either by the linear Basquin model or by the Kohout-Věchet [15] non-linear model. Each load case was covered by at least 5 finished experiments on different load levels, and by one run-out test, which was left unfinished at 10 million cycles. Some experimental test cases were not described well above 750,000 cycles by any of the two mentioned models—see Figure 12 in [14] as regards to the FF033 test case (or see Figure 6 hereafter). This was the main reason why the fatigue strength analysis was covered only at 750,000 cycles in [14]. The newer paper by [16] documents a similar analysis of chosen multiaxial fatigue strength criteria in the limited lifetime region; more precisely at 100,000, 200,000, 500,000 and 750,000 cycles. The tests were performed in compliance with the valid ČSN 42 0362 standard [20]. Every experimental load case is completed by at least one run-out test, for which the specimen did not break even at 10 million cycles. The difference between amplitudes of the dominant stress channel for the run-out specimen and for the last broken specimen with the longest lifetime does not exceed 10 MPa for most cases. The ČSN 42 0363 standard [20] in its paragraph No. 49 recommends choosing this difference in applied stress levels in dependency on the expected fatigue limit of the evaluated tests’ case. The basic number of cycles to determine the fatigue limit is set to 10 million by ČSN 42 0363 for steels.

The value of 10 million cycles is therefore set as the limit of the possible approximation domain. This paper compares results of three approximation methods for the S-N curve data. The first, and the most commonly used method is the Basquin approximation, which should be optimally used only within the experiments with limited lifetime. It is formulated by the following [21]:(1)$$\sigma =\sigma_{f}^{\prime }\cdot {\left(2N\right)}^{b}$$
where $\sigma_{f}^{\prime }$ is the coefficient of fatigue strength, and b is the exponent of fatigue strength. Though this model is more than 100 years old, it presents the part of standards for steel structures (e.g., Eurocode 3, ISO12107).

If the experimental points are selected to be included in the approximation, two tendencies cause some bias. The first one concerns the requirement to include as many experimental points into the regression analysis, as only possible. This requirement ensures that the curve will really correspond to the behavior of material and will not describe the response of only several experimental points. The second tendency to bear in mind are the attempts to omit those data points, which do not conform to the expected material model in the limited lifetime region. For the Basquin model, such data points then can be a part of the S-N curve transition into the quasi-static domain or to the fatigue limit domain, which cannot be approximated by it reasonably well.

A suitable approximation, which can integrate into the model most of experimental data from all those domains, is the Kohout-Věchet regression model ([15], also, K&V hereafter):(2)$$\sigma =a\cdot \left(C\cdot \frac{N+B}{N+C}\right){\;}^{\beta }$$

The non-linear regression analysis demands a certain setup for initial estimates of material parameters a, β, B, C. They can be set based on the previously obtained Basquin regression curve:(3)$$a=2^{b}\cdot \sigma_{f}^{\prime },\;\beta =b,\;B=10^{\left[max_{i}\left(log\sigma_{i}\right)-loga\right]/b},C=10^{\left[min_{i}\left(log\sigma_{i}\right)-loga\right]/b}$$

Thanks to the additional two parameters available in this model in comparison with the Basquin formula, the curve can follow the trend of the S-N data with two bends—one in the transition to the horizontal line at the quasi-static domain, and the other in the transition to the horizontal line at the fatigue limit region. To define better the quasi-static response, the tensile strength at ¼ or ½ cycles can be taken as an additional regression input to other experimental data points. Such inclusion affects, above all, the shape of the curve in the domain of low-cycle fatigue and in the quasi-static domain. However, the Kohout-Věchet model is not suitable for regression of multiple experiments per the outer load levels (load level tests), because the outermost points substantially affect the trend of the curve and its transition to horizontal lines.

To approximate the experimental data, the FF approximation first published in [22] was used. The proposed approximation function is:(4)$$\sigma =\sigma_{0}-\left(\sigma_{0}-\sigma_{C}\right)\cdot \sin\left\{\frac{\pi }{2}\cdot {\left[log\left(4\cdot N_{0}\right)/log\left(4\cdot N_{C}\right)\right]}^{a_{2}}\right\}.$$

This approximation is here used as a one-parametric, where $a_{2}$ is the only fitted parameter and $\sigma_{0}$ is tensile strength. The $\sigma_{C}$ parameter corresponds to the highest stress level, at which the specimen did not break until the lifetime $N_{C}=10^{7}$ cycles. This stress level is in accordance with [19], assumed to correspond to the fatigue limit. Thanks to the use of the sinus function, the approximations are twice bent, which enables to follow the S-N curve trends in both transitions to the horizontal lines. Materials and specimens leading to S-N curve data items that show such S-like trend can thus be suitably modeled by the FF function. On the other hand, the same function limits the use of this formula for lifetimes longer than $N_{C}=10^{7}$ cycles, where the function would start to increase again to higher stresses, which is unlikely for any material. The FF approximation is also not suitable for load level tests. The examples of approximations by the three mentioned formulas can be compared in Figure 6. The functions of the first two formulas (Basquin—Equation (1), Kohout-Věchet—Equation (2)) are shown also outside the interpolation domain of analyzed data, so that their general trends were clearer. Due to the mentioned character of the sinus function, the FF approximation is shown only until $N_{C}=10^{7}$ cycles, and not at higher lifetimes.

The shape of the Kohout-Věchet curve, e.g., for FF033 experiment in repeated torsion (Figure 6), is caused by the model properties, where the transition to the quasi-static region best follows the experimental data items, and it leads to the smallest coefficient of determination $R^{2}$. As only data items related to broken specimens are used for this regression model, the obtained regression curves for most evaluated load cases are limited in their interpolation region only to the lifetimes up to 2 million cycles. Some test cases show the limitation even more stringently, as e.g., the FF033 test case that is usable only up to approx. 900,000 cycles. To get the reasonably set fatigue strengths for each load case, another approximation rather than the Kohout-Věchet model should be applied. To increase the multiaxial fatigue strength analyses reported hereafter, also to the lifetimes of 1, 2, 5 and 10 million cycles, the FF approximation was chosen.

The quality of each of the approximations described in Equations (1), (2) and (4) is compared in Figure 7. The chosen characteristics shown for each load case is the coefficient of determination *R*^2^. In this comparison, the Kohout-Věchet model clearly attains the best results, and the Basquin model and FF model are comparable one to another, but are weaker than the Kohout-Věchet approximation. It should be anyhow reminded, that *R*^2^ parameters are computed on different sets of experimental points—the Basquin curve is regressed only on points in the inclined part of the S-N curve, the Kohout-Věchet curve excludes all run-outs, and only the FF model covers all data points. Logically, its results can be comparably worse to the Kohout-Věchet model in *R*^2^ parameters due to the largest scope of regression inputs.

The parameters of the FF model for each tested load case are summarized in Table 5. To also show the limitation on the scope of usable lifetimes that should be imposed when dealing with the regression curves, the shortest fatigue life obtained experimentally for each load case is documented in Table 5 as *N_min_* parameter.

## 3. Multiaxial Fatigue Strength Criteria

Though the validation program of multiaxial fatigue strength criteria on the presented experimental data concerned 24 calculation methods, this paper presents only 11 of them due to space limits. Mostly, the methods frequently appearing in papers on multiaxial fatigue strength analyses or those implemented in commercial fatigue solvers were selected in addition to some others performing well in other such comparisons [14,23,24,25]. The chosen methods are summarized in Table 6, where each is described by the appropriate reference, by its abbreviation used hereafter and, above all, by its formula.

The multiaxial fatigue strength criteria summarized in Table 6 cover a different approach to multiaxial prediction. The most common solution presently is the critical plane model. If a given hot-spot in which the fatigue crack initiates is evaluated, the critical plane model expects that there is some unique plane on which the stress parameters, when evaluated, result in the highest equivalent stress amplitude, and that this critical plane thus manifests the fatigue response of the whole specimen. From the models selected for the documented validation, the Dan Van criterion [26], the Findley criterion [27], the McDiarmid criterion [28], the Papuga QCP criterion [24,29] and the Papuga PCRN criterion [23,29] belong to this family of criteria. The same concept of one decisive critical plane can use another option to set the critical plane—this is the orientation for which the maximum shear stress range is found on the plane during the load cycle. This solution is here represented by the Matake criterion [30].

Another approach does not look for a critical plane but evaluates the composition of stress parameters on all planes defined by two Euler angles: *φ* and *θ* resulting in its integral mean value to be input into the final equivalent stress amplitude *σ_eq,a_*. These criteria are usually called integral criteria. Two representatives were chosen in this validation—the Liu and Zenner criterion [31] and the Papadopoulos criterion [32]. The latter one processes the projection of the shear stress path on the evaluated plane into a specific direction given by the third angle *χ*. This projection is called resolved shear stress *T.*

In most of these criteria, the processed stress parameters relate to the examined plane—these are normal stress *N* (normal to the plane), and shear stress *C* (lying in the examined plane). All analyses, results of which are described hereafter, were done with the analysis of the stress path parameters via the minimum circumscribed circle concept, as described by Papadopoulos et al. in [25]. It is important to note that there are alternatives to this solution, as discussed by Meggiolaro et al. [33] or by Papuga et al. [34]—e.g., the minimum circumscribed ellipse method or the maximum prismatic hull. The difference in the method of processing the shear stress path would, however, concern only the two load cases of FF003 and FF004 run with the non-zero phase shift (see Table 4).

In addition to critical plane criteria and integral criteria, two more methods remain. There are two criteria—by Crossland [35] and by Sines [36] —that process the history of stress tensor components separated to the description of the stress deviator *J*_2_ and the hydrostatic stress *σ_H_*. To minimize the computation costs, the six stress tensor components can be reduced to five, thanks to the dependency of stress deviator components on its trace. This reduction to five parameters allows to project the load history into 5D Ilyushin’s deviatoric space and the concept of the minimum circumscribed hyperball (or hyperellipsoid) which can be applied to it to provide the amplitude value as the radius of the hyperball.

The last criterion is the recently proposed MMP criterion, which offers the unique simplicity of processing the load history (but only then, when the load cycle is clearly defined). Each of the stress components is treated separately to define its amplitude and mean values, which are then completed into the formula in Equation (12). The validation of this method in [14] showed that despite the simple approach, the results are highly competitive with other commonly used criteria.

The material parameters necessary for each criterion are derived from fatigue strengths in axial loading and in torsion loading. They are reported in Table 7. The stress components that correspond to the fatigue strengths at given *N_x_* cycles are derived from the FF approximation and from the outputs of the FE-analyses. All fatigue strength analyses were run in PragTic fatigue solver [37].

For the validation purposes, equivalent stress amplitude *σ_eq,a_*(*N_x_*) in Equations (6)–(16) was compared with fatigue strengths in fully reversed push–pull loading *p_−1_(N_x_)* or in bending *b_−1_(N_x_)*, which replaced, in the formulas in Equations (6)–(16), the more general fatigue strength in fully reversed axial loading *s_−_*_1_*(N_x_)*. The fatigue strength in fully reversed bending was applied in all multiaxial load cases including the non-zero bending load channel, while all other test cases were analyzed while using the fatigue strength *p_−1_(N_x_)*.

In the validation process, the prediction quality was not assessed directly for each experimental data point, but on fatigue strengths derived from the FF regression curves within whole interpolation domain of each test case. Seven levels of lifetimes (numbers of cycles) were chosen for the analysis in this paper: 0.1, 0.2, 0.5, 1.0, 2.0, 5.0 and 10.0 million cycles.

The relative error between the computed equivalent stress amplitude *σ_eq,a_*(*N_x_*) and the given material response in fully reversed axial fatigue strength *s_−_*_1_(*N_x_*) corresponds to the fatigue index error Δ*FI*(*N_x_*):(5)$$\Delta FI\left(N_{x}\right)=\left(\frac{\sigma_{eq,a}\left(N_{x}\right)-s_{-1}\left(N_{x}\right)}{s_{-1}\left(N_{x}\right)}\right)\cdot 100\%$$

The multiaxial criteria often include the effect of the mean normal stress *N_m_* in their formula, but the effect of the mean shear stress *C_m_* is usually neglected. The exact formulation of the mean stress effect depends on each criterion. Crossland [35] covers the mean stress effect only by $\sigma_{H,max}$, which corresponds to the maximum hydrostatic stress during the loading cycle. Focusing on the mean normal stress only is probably historically caused by Sines [36], who postulated on the basis of gathered experimental data, that the mean shear stress has only limited effect on the fatigue limit, unless it exceeds the value of yield stress in shear. Papadopoulos et al. [25] applied this assumption as the requirement for assessing the suitability of various fatigue strength methods, and their work affected many researchers who did not object to it.

**Table 6 materials-14-00116-t006:** Formulas of methods used within the validation program. Parameters used: *s_−1_*—fatigue strength in fully reversed axial loading, *S_u_*—tensile strength, $J_{2}$—second invariant of the deviatoric stress tensor, *σ**_H_*—hydrostatic stress, *C*—shear stress on the examined plane, *N*—normal stress on the examined plane, *θ*, *φ*—Euler angles defining the orientation of the examined plane, *T*—resolved shear stress (shear stress projection into a direction described by *χ* angle), *w*—Walker’s mean stress effect parameter. Indexes *a* and *m* designate the amplitude and mean values, respectively. Formulas for computing *a*, *b*, *c*, *d* parameters are provided in Table 7.

Criterion	Abbrev.	Formulation of the Equivalent Stress Amplitude	Equation
Crossland	CROSS	$\sigma_{eq,a}=a_{C}\cdot {\left(\sqrt{J_{2}}\right)}_{a}+b_{C}\cdot \sigma_{H,max}$	(6)
Dang Van	DV	$\sigma_{eq,a}=\underset{\varphi ,\theta }{\max}\left(a_{DV}\cdot C_{a}+b_{DV}\cdot \sigma_{H,max}\right)$	(7)
Findley	FIN	$\sigma_{eq,a}=\underset{\varphi ,\theta }{\max}\left(a_{F}\cdot C_{a}+b_{F}\cdot N_{max}\right)$	(8)
Liu-Zenner	LZ	$\sigma_{eq,a}=\sqrt{\int_{\varphi =0}^{2\pi }\int_{\theta =0}^{\pi }\left[a_{L}C_{a}^{2}\left(1+c_{L}C_{m}^{2}\right)+b_{L}N_{a}^{2}\left(1+d_{L}N_{m}^{}\right)\right]sin\theta \;d\theta \;d\varphi }$	(9)
Matake	MATA	$\sigma_{eq,a}=a_{M}\cdot C_{a}+b_{M}\cdot N_{max}$	(10)
McDiarmid	MCD	$\sigma_{eq,a}=\underset{\varphi ,\theta }{\max}\left(\frac{s_{-1}}{t_{AB}}\cdot C_{a}+\frac{s_{-1}}{2S_{u}}\cdot N_{max}\right)$	(11)
Manson-McKnight-Papuga	MMP	$\sigma_{eq,a}=\sigma_{aP}^{w}\cdot {\left(\sigma_{aP}+\beta_{P}\cdot \sigma_{mP}\right)}^{1-w}\leq s_{-1}$ $\sigma_{aP}=\sqrt{\frac{1}{2}\left[\begin{array}{c} {\left(\sigma_{x,a}-\sigma_{y,a}\right)}^{2}+{\left(\sigma_{y,a}-\sigma_{z,a}\right)}^{2}+{\left(\sigma_{z,a}-\sigma_{x,a}\right)}^{2}\\ +2\cdot \kappa^{2}\left(\sigma_{xy,a}^{2}+\sigma_{yz,a}^{2}+\sigma_{zx,a}^{2}\right) \end{array}\right]}$ $\sigma_{mP}=\sqrt{\frac{1}{2}\left[\begin{array}{c} {\left(\sigma_{x,m}-\sigma_{y,m}\right)}^{2}+{\left(\sigma_{y,m}-\sigma_{z,m}\right)}^{2}+{\left(\sigma_{z,m}-\sigma_{x,m}\right)}^{2}\\ +2\cdot X_{m}^{2}\left(\sigma_{xy,m}^{2}+\sigma_{yz,m}^{2}+\sigma_{zx,m}^{2}\right) \end{array}\right]}$	(12)
Papadopoulos	PAPADO	$\sigma_{eq,a}=\sqrt{a_{P}\cdot T_{a}^{2}}+b_{P}\cdot \sigma_{H,max}$ $\sqrt{T_{a}^{2}}=\sqrt{\frac{5}{8\pi^{2}}\int_{\varphi =0}^{2\pi }\int_{\theta =0}^{\pi }\int_{\chi =0}^{2\pi }{\left(T_{a}\left(\varphi ,\theta ,\chi \right)\right)}^{2}d\chi \;sin\theta \;d\theta \;d\varphi }$	(13)
Papuga QCP	QCP	$\sigma_{eq,a}=\underset{\varphi ,\theta }{\max}\sqrt{a_{Q}\cdot C_{a}\left(C_{a}+c_{Q}\cdot C_{m}\right)+b_{Q}\cdot N_{a}\left(N_{a}+d_{Q}\cdot N_{m}\right)}$	(14)
Papuga PCRN	PCRN	$\sigma_{eq,a}=\underset{\varphi ,\theta }{\max}\sqrt{a_{I}\cdot C_{a}\left(C_{a}+c_{I}\cdot C_{m}\right)+b_{I}\cdot \sqrt{N_{a}\left(N_{a}+d_{I}\cdot N_{m}\right)}}$	(15)
Sines	SINES	$\sigma_{eq,a}=a_{S}\cdot {\left(\sqrt{J_{2}}\right)}_{a}+b_{S}\cdot \sigma_{H,m}$	(16)

**Table 7 materials-14-00116-t007:** Material parameters of individual criteria and their limitations. *s_−1_*—fatigue strength in fully reversed axial loading, *t_−1_*—fatigue strength in fully reversed torsion, *t_0_*—fatigue strength in repeated torsion (maximum stress of the cycle), *s_0_*—fatigue strength in repeated axial loading (maximum stress of the cycle), *σ**_1_* is maximum principal stress, *σ**_3_* is minimum principal stress, *κ* is ratio of fatigue strengths in fully reversed loadings (*s_−_*_1_/*t*_−1_), *κ_0_* is ratio of fatigue strengths in repeated loadings (*s*_0_/*t*_0_).

Criterion	Formula	Equation
Crossland	$a_{C}=\frac{s_{-1}}{t_{-1}},\;b_{C}=3-\sqrt{3}\cdot \frac{s_{-1}}{t_{-1}}$	(17)
Dang Van	$a_{DV}=\frac{s_{-1}}{t_{-1}},\;b_{DV}=3-\frac{3}{2}\cdot \frac{s_{-1}}{t_{-1}}$	(18)
Findley	$a_{F}=2\cdot \sqrt{\frac{s_{-1}}{t_{-1}}-1},\;b_{F}=2-\frac{s_{-1}}{t_{-1}}$	(19)
Liu-Zenner	$a_{L}=\frac{3}{2}\left[3{\left(\frac{s_{-1}}{t_{-1}}\right)}^{2}-4\;\right],\;c_{L}=\frac{28}{3a_{L}\cdot t_{0}^{4}}\left[s_{-1}^{2}-{\left(\frac{\frac{s_{-1}}{t_{-1}}\cdot t_{0}}{2}\right)}^{2}\right]$ $b_{L}=3\left[3-{\left(\frac{s_{-1}}{t_{-1}}\right)}^{2}\right],\;d_{L}=\frac{28}{15b_{L}\cdot s_{0}}\left[{\left(\frac{2s_{-1}}{s_{0}}\right)}^{2}-\frac{4}{21}c_{L}\cdot a_{L}\cdot {\left(\frac{s_{0}}{2}\right)}^{2}-1\right]$	(20)
Matake	$a_{M}=\frac{s_{-1}}{t_{-1}}\;,\;b_{M}=2-\frac{s_{-1}}{t_{-1}}$	(21)
Manson-McKnight-Papuga	$w=\frac{log\frac{s_{0}}{s_{-1}}}{log2},\;X_{m}=2\cdot \kappa \cdot \left[{\left(\frac{2\cdot t_{-1}}{t_{0}}\right)}^{\frac{1}{1-w}}-1\right]$ $\left|\sigma_{1,max}\right|\geq \left|\sigma_{3,min}\right|:\beta_{P}=\frac{\sigma_{1,max}}{\sigma_{1,max}-\sigma_{3,min}}$ $\left|\sigma_{1,max}\right|<\left|\sigma_{3,min}\right|:\beta_{P}=\frac{\sigma_{1,min}}{\sigma_{1,max}-\sigma_{3,min}}$	(22)
Papadopoulos	$a_{P}={\left(\frac{s_{-1}}{t_{-1}}\right)}^{2},\;b_{P}=3-\sqrt{3}\cdot \frac{s_{-1}}{t_{-1}}$	(23)
Papuga QCP	$a_{QCP}=\kappa^{2}\;,\;\;\;c_{QCP}=\frac{4\cdot s_{-1}^{2}}{b\cdot t_{0}^{2}}-1$ $\kappa <\sqrt{2}\;:\;\;\;\;b_{QCP}=1\;,d_{QCP}=\frac{4\cdot s_{-1}^{2}}{b_{QCP}\cdot s_{0}^{2}}-1$ $\kappa \geq \sqrt{2}\;:\;\;\;\;b_{QCP}=\kappa^{2}-\frac{\kappa^{4}}{4}\;,d_{QCP}=\frac{4\cdot s_{-1}^{2}}{b_{QCP}\cdot t_{0}^{2}}\cdot \left(1-\frac{\kappa^{2}}{4}\;\right)-1$	(24)
Papuga PCRN	$1\leq \kappa \;\left(\kappa_{0}\right)<\sqrt{\frac{4}{3}}\;:\;a_{I}=\frac{\kappa^{2}}{2}+\frac{\sqrt{\kappa^{4}-\kappa^{2}}}{2}\;,\;b_{I}=s_{-1}$ $c_{I}=\frac{2s_{-1}^{2}}{a_{I}\cdot t_{0}^{2}}\cdot \left(1+\sqrt{1-\frac{1}{\kappa_{0}{}^{2}}}\right)-1,\;d_{I}={\left(\frac{2s_{-1}^{2}}{b_{I}\cdot s_{0}}\right)}^{2}-1$ $\kappa \;\left(\kappa_{0}\right)\geq \sqrt{\frac{4}{3}}:a_{I}={\left(\frac{4\cdot \kappa^{2}}{4+\kappa^{2}}\right)}^{2},\;b_{I}=8\cdot s_{-1}\cdot \kappa^{2}\cdot \frac{4-\kappa^{2}}{{\left(4+\kappa^{2}\right)}^{2}},\;c_{I}=\frac{z}{a_{I}}-1,\;d_{I}=\frac{z}{b_{I}{}^{2}}\cdot \left(4\cdot s_{-1}^{2}-z\cdot t_{0}^{2}\right)-1,\;z={\left[\frac{8\cdot \kappa_{0}\cdot s_{-1}}{t_{0}\cdot \left(4+\kappa_{0}{}^{2}\right)}\;\right]}^{2}$	(25)
Sines	$a_{S}=\frac{s_{-1}}{t_{-1}},\;b_{S}=6\cdot \frac{s_{-1}}{s_{0}}-\sqrt{3}\cdot \frac{s_{-1}}{t_{-1}}$	(26)

Experiments and references cited by Papuga and Halama in [35] document that including the mean shear stress into the criterion can improve the prediction quality, which is proven on PCRN and QCP criteria. Acceptance of this effect and its inclusion into the formulas imposes another requirement on material parameters to be used—the S-N curve in repeated torsion is necessary. In the test set presented here, the FF033 test case performed on S1 configuration of specimens is available for these smaller specimens. To get the appropriate fatigue strengths in repeated torsion, also for bigger S2 and S3 specimens, the formula proposed by Zenner et al. [28] is used:(27)$$\frac{4\cdot t_{-1}}{t_{0}}-\frac{2\cdot s_{-1}}{s_{0}}=1$$

The same formula was used by Papuga in [35]. Papuga described that the validation of the formula in Equation (27) for test sets for which all four fatigue strengths were available showed that the relative error of the estimated fatigue strength in repeated torsion did not exceed 5%.

## 4. Discussion of Results

Results of chosen multiaxial fatigue strength estimation methods from Equations (6)–(16) are statistically processed and provided to the reader in Table 8, Table 9 and Table 10. Table 8 describes the mean Δ*FI* fatigue index errors. It also summarizes the sum of Δ*FI* squares over all evaluated lifetimes, which nicely documents the overall prediction quality. To support a quicker evaluation of this output, the conditional formatting of this parameter over all validated methods is shown, with the red color marking the worst results and the green color highlighting the best results. The same system of conditional formatting is also used in Table 9 and Table 10. The mean Δ*FI* errors are provided at each evaluated lifetime and also for all of them together to document the overall trend and the potential deviations from it for individual lifetimes. The results gathered in Table 8 highlight the good prediction quality of PCRN, MMP and QCP formulas, and it also clearly shows the much worse results of SINES, CROSS, PAPADO and FINDLEY criteria.

A quite important question is whether the individual criteria will show some systematic change of the Δ*FI* output at different lifetimes. Table 8 and Table 9 show that there are changes for different lifetime levels, but the values remain quite stable—the typical range of values of 2% can be found for most cases. This confirms the expectation by Karolczuk et al. [36] that the fatigue strengths at the evaluated lifetime (and not for some hypothetical fatigue limit) should be used for computing the material parameters of individual multiaxial fatigue criteria. The relative insensitiveness of the Δ*FI* level to *N_x_* fatigue life, at which it is computed, demonstrates also that the regression by the FF function did not affect the output in a negative way.

Table 9 states the sample standard deviations of Δ*FI*, both at individual lifetimes and over all of them, where the conditional formatting is again used. The evaluation of individual methods do not differ substantially from the conclusion made for Table 8.

The smallest sample standard deviation can be detected by PCRN, MMP and QCP methods. Its highest value is the result of applying SINES, PAPADO and CROSS methods. If the overall prediction quality is compared over all evaluated lifetimes, the PCRN, QCP and MMP methods provide better output than other validated methods. If the sample standard deviation of PCRN is compared with the Dang Van method (DV), which is the most often used solution in the engineering practice, it can be noted that the sample standard deviation of Δ*FI* for the PCRN criterion is three times smaller than the sample standard deviation of the DV criterion.

Table 10 shows minimum and maximum Δ*FI* values and its variation range. It can be concluded that even this parameter results in a very similar ranking of individual fatigue strength criteria as found in Table 8 and Table 9. Histograms of Δ*FI* occurrence in all experiments and at all evaluated lifetimes are depicted in Figure 8 for all validated multiaxial fatigue strength criteria.

## 5. Conclusions

The analysis presented in this paper focuses on validating 11 different multiaxial fatigue strength criteria on the own test set composed of 34 S-N curves describing the fatigue response for different multiaxial load cases. The S-N curves were approximated on the experimental data based on the FF regression function. All test cases relate to hollow specimens manufactured from a single melt of the ČSN 41 1523 structural steel equivalent to S355JR.

Eleven chosen fatigue strength criteria comprise commonly used criteria and some new criteria, which were recently proven to provide good prediction results. For each load case, the validation is realized on fatigue strengths retrieved from the FF approximation at seven different lifetimes between 0.1 and 10 million cycles. The results are assessed based on the defined fatigue index error Δ*FI* and its statistical processing over all evaluated lifetimes and checked load cases. The mean value of Δ*FI*, the sum of its squares, sample standard deviation or its variation range are assessed. Based on the comparison for various evaluated multiaxial criteria, for the tested material and for tested load cases, it can be concluded:PCRN, QCP and MMP methods result in the best prediction quality, and their sample standard deviation is multiple times lower than the sample standard deviation of the Dang Van method, though the Dang Van method is commonly used in the engineering practice.In the range between 0.1 and 10 million cycles, the prediction quality is stable, and the variability of Δ*FI* over this interval is negligible if compared with the overall prediction scatter. This holds true only then, when the complete calculation of the equivalent stress amplitude and of the material parameters specific to each multiaxial fatigue strength criterion, as described in Table 6 and Table 7, is done at the same fatigue life.

## Figures and Tables

**Figure 1 materials-14-00116-f001:**
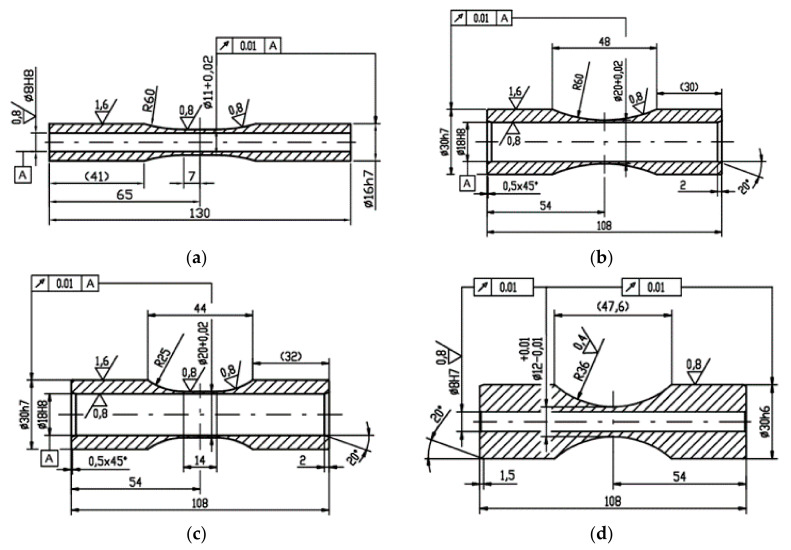
Drawings of specimens used in this campaign: (**a**) S1—top left, (**b**) S2—top right, (**c**) S3—bottom left, (**d**) S4—bottom right.

**Figure 2 materials-14-00116-f002:**
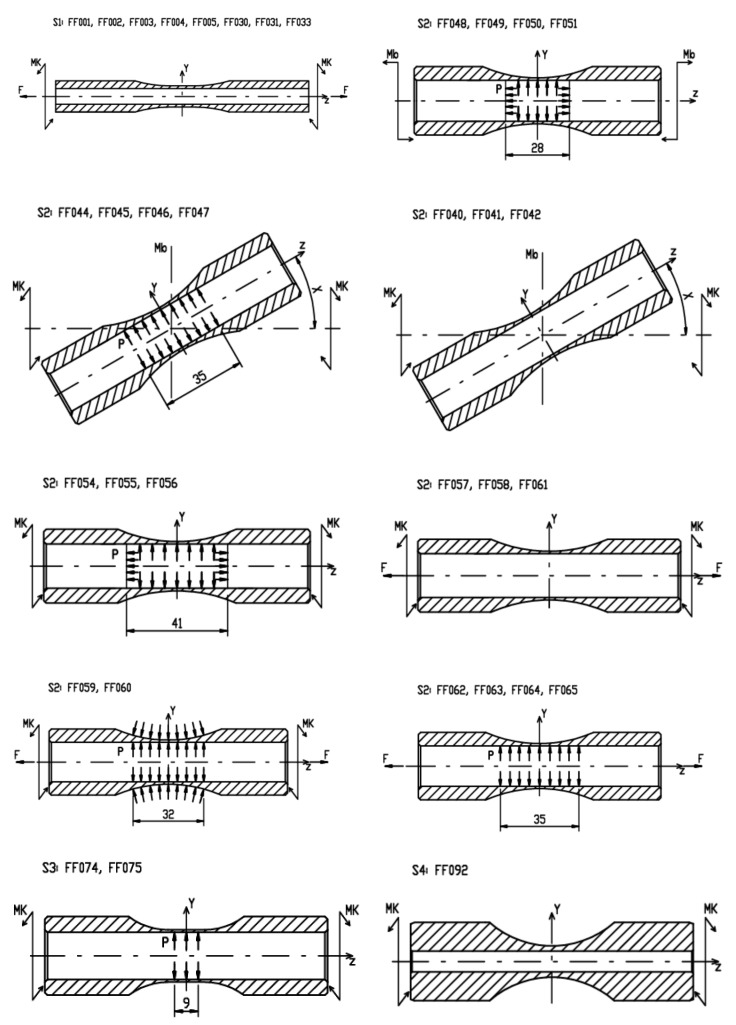
Overview of various setups of experiments. *F*—push/pull, *Mb*—bending moment, *Mk*—torque and *P*—pressure.

**Figure 3 materials-14-00116-f003:**
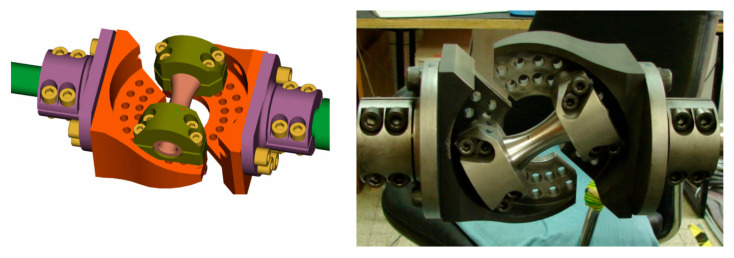
Setup of the experiments for load cases of torsion and bending on the torque controlled FF044 (right) and bending load case FF040 (left), both using S2 specimens.

**Figure 4 materials-14-00116-f004:**
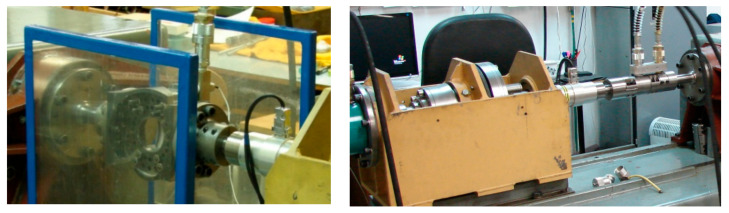
The load case of bending and pressurizing with the pressure chamber and the pressure sensor—FF046 load case (left). The setup of the test case with the pressure chamber inducing inner and outer pressure—FF059 load case (right).

**Figure 5 materials-14-00116-f005:**
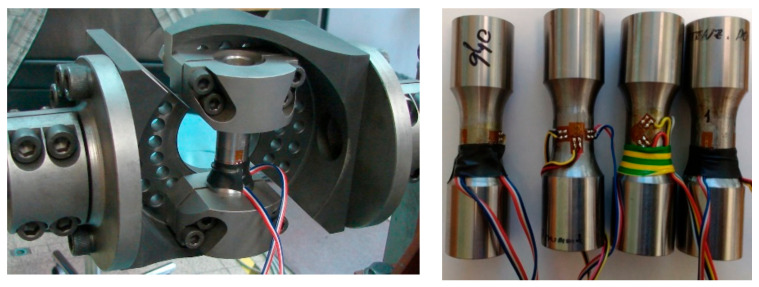
Examples of the load setup validations taken to ensure the applied stress are conforming to the expectations.

**Figure 6 materials-14-00116-f006:**
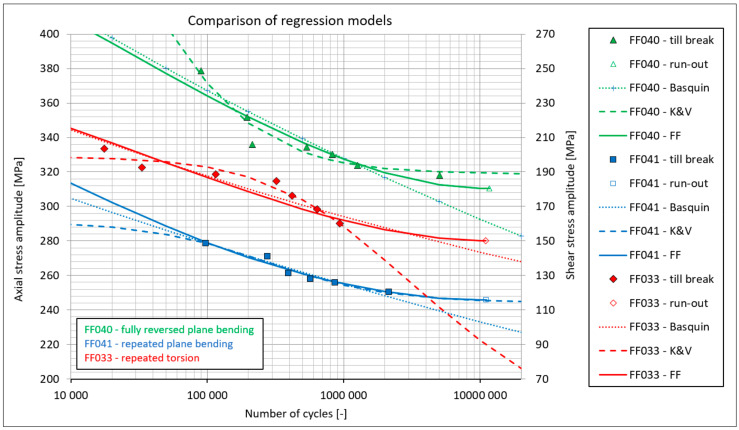
Comparison of the three approximation models for three different load cases.

**Figure 7 materials-14-00116-f007:**
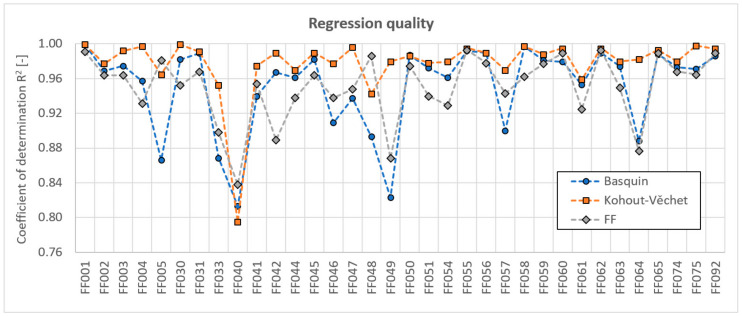
Comparison of the coefficient of determination *R*^2^ for 3 evaluated regression formulas and individual load cases.

**Figure 8 materials-14-00116-f008:**
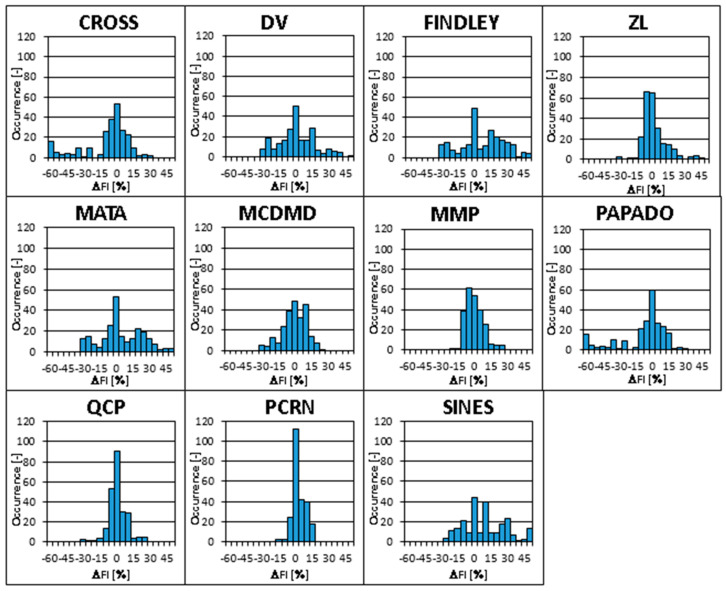
Histograms of Δ*FI* errors for all 34 test cases, 7 evaluated lifetimes and all 11 multiaxial fatigue strength criteria validated according to Table 6.

**Table 1 materials-14-00116-t001:** Static material parameters and chemical composition of the studied material.

Designation	Tensile Strength [MPa]	Tensile Yield Stress [MPa]	Elongation at Fracture [%]	Reduction of Area at Fracture [%]	True Fracture Strength in Torsion [MPa]
ČSN 41 1523	560	400	31.1	74.0	516.6

**Table 2 materials-14-00116-t002:** Chemical composition of the studied material.

Chemical Composition:					
C [%]	Mn [%]	Si [%]	P [%]	S [%]	Cu [%]
0.18	1.38	0.4	0.018	0.006	0.05

**Table 3 materials-14-00116-t003:** Summary of all load cases. Abbreviations used: Ten—tension, To—torsion, RP—pressurized, PV—pressure vessel mode, PB—plane bending, A—amplitude, M—mean value, *D*—outer diameter, *d*—inner diameter.

Mark	Specimen Type	Machine No. (Frequency [Hz])	Load Combination	Measured Diameters		Input Load Channels		
				*D* [mm]	*d* [mm]	Ten PB	RP PV	To
FF001	S1	1 (10)	Ten + To	10.95	8.02	A		A
FF002	S1	1 (10)	Ten + To	10.95	8.02	A		A
FF003	S1	1 (10)	Ten + To	10.95	8.02	A		A
FF004	S1	1 (10)	Ten + To	10.95	8.02	A		A
FF005	S1	1 (10)	Ten + To	10.95	8.02	A, M		A, M
FF030	S1	1 (10)	Ten	10.95	8.02	A		
FF031	S1	1 (10)	Ten	10.95	8.02	A, M		
FF033	S1	1 (10)	To	10.95	8.02			A, M
FF040	S2	2 (25)	PB	19.99	18.05	A		
FF041	S2	3 (10)	PB	19.96	18.06	A, M		
FF042	S2	2 (25)	To	19.99	18.05			A
FF044	S2	2 (25)	PB + To	19.99	18.05	A		A
FF045	S2	2 (25)	PB + To	19.99	18.05	A		A
FF046	S2	2 (25)	PB + RP	20.00	18.03	A	M	
FF047	S2	2 (25)	PB + RP	20.02	18.02	A	M	
FF048	S2	4 (20)	PB + PV	19.93	18.05	A, M	M	
FF049	S2	4 (20)	PB + PV	19.94	18.05	A, M	M	
FF050	S2	4 (20)	PB + PV	19.94	18.05	A, M	M	
FF051	S2	4 (20)	PB + PV	19.92	18.05	A, M	M	
FF054	S2	2 (25)	To + PV	19.97	18.05		M	A
FF055	S2	2 (25)	To + PV	19.94	18.09		M	A
FF056	S2	2 (25)	To + PV	19.94	18.07		M	A
FF057	S2	2 (25)	To + Ten	19.99	18.05	M		A
FF058	S2	2 (25)	To + Ten	19.99	18.05	M		A
FF059	S2	2 (25)	To + Ten + RP	20.04	18.25	M	M	A
FF060	S2	2 (25)	To + Ten + RP	19.96	18.29		M	A
FF061	S2	3 (4)	To + Ten	19.99	18.05	A		A
FF062	S2	4 (20)	Ten + RP	20.00	18.02	A	M	
FF063	S2	4 (20)	Ten + RP	20.01	18.02	A	M	
FF064	S2	4 (20)	Ten + RP	20.01	18.02	A, M	M	
FF065	S2	4 (20)	Ten + RP	20.03	18.02	A, M	M	
FF074	S3	2 (25)	To + RP	20.00	18.05		M	A
FF075	S3	2 (25)	To + RP	20.00	18.05		M	A
FF092	S4	2 (25)	To	12.00	8.00			A

**Table 4 materials-14-00116-t004:** Summary of all load cases as regards to the range of applied forces, moments and pressures; also, the phase shift *φ_at_* between the axial and torsion load signals is stated.

Load Case	Axial Force [kN]		Torque [Nm]		Bending Moment [Nm]		Phase Shift [°] *φ**_at_*	Pressure [MPa]
	*F_a_*	*F_m_*	*Mk_a_*	*Mk_m_*	*Mb_a_*	*Mb_m_*		
	From–To	From–To	From–To	From–To	From–To	From–To		*P_m_*
FF001	4.34	0	34–25	0	0	0	0	0
FF002	8.5	0	24.5–16	0	0	0	0	0
FF003	4.34	0	34–26.15	0	0	0	90	0
FF004	8.5	0	33–23	0	0	0	90	0
FF005	5	5	24.1–17.5	24.1–17.5	0	0	0	0
FF030	12.85–10.4	0	0	0	0	0	0	0
FF031	9.5–8.2	9.5–8.2	0	0	0	0	0	0
FF033	0	0	37.5–27.7	37.5–27.7	0	0	0	0
FF040	0	0	0	0	99.6–81.6	0	0	0
FF041	0	0	0	0	73.3–64.7	73.3–64.7	0	0
FF042	0	0	94.5–83.9	0	0	0	0	0
FF044	0	0	81.1–63.7	0	60.5–47.5	0	0	0
FF045	0	0	42.2–33	0	94.4–73.7	0	0	0
FF046	0	0	0	0	90.1–76.5	0	0	23.3
FF047	0	0	0	0	88.0–74.0	0	0	36.0
FF048	0	0	0	0	65.6–59.6	65.6–59.6	0	10.5
FF049	0	0	0	0	67.7–57.6	67.7–57.6	0	20.0
FF050	0	0	0	0	68.5–56.5	68.5–56.5	0	30.0
FF051	0	0	0	0	66.7–54.1	66.7–54.1	0	40.0
FF054	0	0	93.0–76.2	0	0	0	0	15.0
FF055	0	0	95.0–78.0	0	0	0	0	10.0
FF056	0	0	75.5–62.8	0	0	0	0	20.2
FF057	0	14	98.8–71.6	0	0	0	0	0
FF058	0	10.2	100.6–76.6	0	0	0	0	0
FF059	0	0	91.0–77.5	0	0	0	0	40.0
FF060	0	5.9	84.7–77.0	0	0	0	0	40.0
FF061	16.6–12.3	0	27.1–20.1	0	0	0	0	0
FF062	17.0–14.6	0	0	0	0	0	0	20.0
FF063	16.5–13.0	0	0	0	0	0	0	40.0
FF064	7.2–6.3	7.2–6.3	0	0	0	0	0	20.0
FF065	7.3–6.4	7.3–6.4	0	0	0	0	0	40.0
FF074	0	0	87.6–74.9	0	0	0	0	13.0
FF075	0	0	85.7–48.2	0	0	0	0	27.0
FF092	0	0	55.7–45.2	0	0	0	0	0

**Table 5 materials-14-00116-t005:** Parameters of the FF approximations (Equation (4)) for each load case, including also the lowest measured experimental lifetime *N_min_*, which together with the run-out level at 10,000,000 cycles define the complete interpolation region of each regression curve.

Mark	$a_{2}$ [-]	$\sigma_{0}$ [MPa]	$\sigma_{C}$ [MPa]	$N_{0}$ [-]	$N_{min}$ [-]
FF001	1.366	405.5	135.6	0.25	59,663
FF002	1.6989	319.9	87.0	0.25	101,102
FF003	3.108	215.2	141.8	0.25	38,206
FF004	3.061	206.1	124.7	0.25	19,604
FF005	3.551	153.6	95.3	0.25	31,769
FF030	0.92	734.1	238.9	0.25	28,562
FF031	1.388	327.0	187.3	0.25	29,150
FF033	2.153	280.9	150.2	0.25	17,553
FF040	2.001	520.5	310.3	0.25	89,700
FF041	1.35	480.4	245.8	0.25	97,324
FF042	2.349	230.1	159.5	0.25	92,200
FF044	2.369	217.8	121.1	0.25	56,102
FF045	2.09	107.8	62.7	0.25	33,100
FF046	0.911	837.6	286.9	0.25	81,900
FF047	0.873	805.4	273.4	0.25	66,537
FF048	1.163	464.3	234.5	0.25	123,147
FF049	0.512	1362.4	225.2	0.25	79,393
FF050	1.304	540.5	220.8	0.25	72,732
FF051	1.207	610.6	214.1	0.25	75,425
FF054	2.509	247.4	146.6	0.25	96,540
FF055	2.729	231.7	155.3	0.25	51,330
FF056	8.14	153.2	123.9	0.25	175,000
FF057	2.135	233.4	136.1	0.25	8886
FF058	2.404	240.7	145.7	0.25	18,210
FF059	5.815	191.3	157.1	0.25	82,310
FF060	5.774	189.2	167.2	0.25	97,500
FF061	6.216	294.7	212.7	0.25	45,723
FF062	1.818	402.2	244.9	0.25	69,061
FF063	0.828	661.9	217.0	0.25	15,656
FF064	1.656	367.4	208.6	0.25	135,447
FF065	6.263	252.3	210.6	0.25	220,139
FF074	2.996	204.4	142.5	0.25	120,600
FF075	2.588	304.0	91.6	0.25	80,310
FF092	2.962	234.7	166.2	0.25	30,800

**Table 8 materials-14-00116-t008:** Results of the Δ*FI* statistics for 11 methods at different *N_x_*—sum of squares and mean values.

Comp. Method	Sum of Δ*FI* Squares	Mean Value of Δ*FI* for *N_x_*							
		All	100,000	200,000	500,000	1 × 10^6^	2 × 10^6^	5 × 10^6^	1 × 10^7^
CROSS	1334%	−9.1%	−9.2%	−8.9%	−8.5%	−8.2%	−9.1%	−9.8%	−10.1%
DV	841%	2.4%	1.9%	2.2%	2.8%	3.2%	2.6%	2.1%	1.8%
FINDLEY	1134%	7.8%	7.0%	7.7%	8.5%	8.7%	8.1%	7.5%	7.1%
LZ	295%	2.0%	3.6%	3.3%	2.8%	1.7%	1.5%	0.7%	0.2%
MATA	864%	4.5%	4.0%	4.5%	5.1%	5.2%	4.7%	4.1%	3.8%
MCDMD	279%	−0.2%	2.7%	1.7%	0.5%	−0.9%	−1.1%	−1.9%	−2.2%
MMP	148%	0.3%	1.6%	1.4%	1.0%	0.0%	−0.1%	−0.8%	−1.2%
PAPADO	1341%	−8.3%	−8.6%	−8.1%	−7.7%	−7.4%	−8.2%	−8.8%	−9.1%
QCP	165%	0.7%	2.3%	2.0%	1.5%	0.4%	0.3%	−0.4%	−0.8%
PCRN	98%	2.8%	4.0%	3.9%	3.4%	2.6%	2.5%	1.9%	1.6%
SINES	1561%	11.7%	13.1%	13.0%	12.5%	11.6%	11.3%	10.5%	10.0%

**Table 9 materials-14-00116-t009:** Results of the Δ*FI* statistics for 11 methods at different *Nx*—sample standard deviations.

Comp. Method	Sample Standard Deviation of Δ*FI* for *Nx*							
	All	100,000	200,000	500,000	1 × 10^6^	2 × 10^6^	5 × 10^6^	1 × 10^7^
CROSS	21.9%	22.8%	22.8%	22.8%	22.0%	22.1%	21.5%	21.2%
DV	18.7%	18.2%	18.9%	19.5%	19.4%	19.2%	18.7%	18.4%
FINDLEY	20.4%	20.1%	20.7%	21.3%	21.3%	21.0%	20.4%	20.0%
LZ	11.0%	10.9%	11.0%	11.1%	10.8%	11.3%	11.3%	11.1%
MATA	18.6%	18.9%	19.2%	19.5%	19.1%	18.8%	18.2%	17.8%
MCDMD	11.1%	11.1%	10.9%	10.8%	10.5%	10.8%	10.9%	10.9%
MMP	7.9%	8.1%	8.0%	8.0%	7.6%	8.0%	8.0%	7.9%
PAPADO	22.3%	23.2%	23.2%	23.2%	22.4%	22.5%	22.0%	21.6%
QCP	8.3%	7.1%	7.4%	7.9%	8.3%	8.8%	9.2%	9.3%
PCRN	5.8%	5.4%	5.4%	5.6%	5.6%	6.0%	6.2%	6.3%
SINES	22.8%	22.7%	23.1%	23.5%	23.2%	23.4%	23.1%	22.6%

**Table 10 materials-14-00116-t010:** Results of the Δ*FI* statistics for 11 methods at different *N_x_*—minimum and maximum values and the variation range.

Comp. Method	Minimum	Maximum	Variation Range Δ*FI*
CROSS	−70.6%	30.5%	101.1%
DV	−30.1%	65.2%	95.3%
FINDLEY	−30.4%	58.4%	88.8%
LZ	−30.2%	42.8%	73.0%
MATA	−30.4%	49.4%	79.8%
MCDMD	−28.3%	22.7%	51.1%
MMP	−18.5%	25.2%	43.6%
PAPADO	−70.6%	30.5%	101.1%
QCP	−31.8%	27.5%	59.3%
PCRN	−16.5%	17.0%	33.5%
SINES	−25.0%	92.7%	117.6%

## Data Availability

The data presented in this study are available on request from the corresponding author. The data are not publicly available due to their complexity.

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
