# Peer review of "Validation of Multiaxial Fatigue Strength Criteria on Specimens from Structural Steel in the High-Cycle Fatigue Region"

_materials, 2020, doi:10.3390/ma14010116_

Round 1

Reviewer 1 Report

The main objective of this work seems to be to compare the prediction and experimental fatigue lifetimes obtained from different multiaxial fatigue criteria applied on different multiaxial loading scenarios. To do that,  eleven different fatigue failure criteria were used to perform predictions of failure for 34 configurations of applied loads. After that, the results obtained from that predictions were compared with experimental results.

The paper is mainly focused on the description of the geometries, applied loads scenarios, fitting of the S-N curves obtained, description of the calculation methods (fatigue criteria) and discussion of the fatigue results. Nevertheless, all parts enumerated above were strongly resumed, and not enough well explained in the paper. In general terms, the paper can not be understood adequately without reading (at the same time) reference [13] from the same authors, which can be considered a limitation of the paper. Furthermore, the structure of the paper and the organization of the information could be improved (see some suggestions on particular comments).   

In conclusion, the reviewer's option is that, although the paper's topic is interesting and the results can be an excellent contribution to the state of the art because they can be useful for predicting fatigue failures, the paper should be considerably improved before publication. Thus, the reviewer suggests publishing the paper after a major revision that should respond to the following comments/suggestions.

Comment 1

Title: Consider to modify the title of the paper. There is not a validation on the paper, only a comparison under different fatigue criteria.

"Validation of multiaxial fatigue strength criteria on  specimens from structural steel in the high-cycle fatigue region"

Comment 2

First of all, the paper presents an introduction to the problem, which includes a brief motivation related to the necessity of performing various load combinations of push-pull and torsion tests to measure the quality of a multiaxial fatigue criterion, and the credibility of the results available in the literature (lines 29 to 52). After that, the authors start to describe the motivation, objectives, results and conclusions of reference [13], which forces the reader to read that reference in detail in order to understand the scope, motivation and objectives of the present paper. Although it is interesting to relate the present paper with the results obtained on reference [13], the actual paper should not be presented as a "simple extension" of the previous paper, which seems to be the intention of the authors (lines 53 to 70).

The main objectives of the paper must be highlighted in the introduction, and the way to reach these objectives could be resumed on the introduction. Additionally, the authors could include a short paragraph describing the paper organization.

Comment 3

The second section of the paper (2. Experiments) must be improved. Different information is mixed in the same section, and too many calls to reference [13] are included.

The reviewer suggests to divide the information into different subsections and enlarge the information provided (avoiding to force the reader to consult ref [13](when it is possible)) Example:

2.1 Material (lines 75 to 80)

2.2 Geometry (lines 72 to 75 and figure 1) à consider to include S1, S2 and S3 figures on the current paper.

2.3 Loading Scenarios (lines 83 to  83 and 101 to 113;  Figure 2 and Table 3) à Consider to homogenize the description of the load scenarios (description of new experimental load cases (lines 87 to 93) differs substantially to the description of previous load cases (lines 101 to 103). Try to explain the load cases better; as an example, there is no clear difference between the different load scenarios FF048; FF04; FF050 and FF051 (because all of them correspond to the same figure (see figure 2)).

2.4 Testing procedure (lines 106 to 120) à Please, try to divide the actual paragraph into different paragraphs to describe the different load conditions and machines used.

2.5 Obtention of local stresses (lines 133 to 143) à please include some graphics representing examples of the stress distributions obtained.

2.6 Fitting the S-N curves (lines 160 to 199) à  Describe the 3 alternatives considered to fit the S-N curves obtained

2.7 Discussión the results obtained on the experimental programme à Describe the results and discuss the main conclusions obtained in this section (including comparison with previous works (lines 144 to 157).

Comment 4

"The approximation quality of the three mentioned formulas can be compared in Figs. 5-7." à What is the meaning of "approximation quality"? Can be the quality of formulas measured in an objective way?

"The functions of the first two formulas (Basquin – Eq. (1), Kohout-VÄ›chet – Eq. (2)) are shown also outside the interpolation domain of analyzed data, so that their trends were manifested better." à Are the trends better because the formulas allow the extrapolation? Consider rewriting the sentence because all types of extrapolations could not be considered "manifested better".

"Due to the reason mentioned previously, the FF approximation is shown only until ?? = 107 cycles." à the reason of shown only to 10^7 is maybe related to Eq. 3, not to the previous phrase.

"The application of other regression models (Stromeyerova [21], Palmgren [22], Gecks-Ochs [23]) was also evaluated but none of them provided better results as regards the approximation quality." à It is not demonstrated in the paper, nor justified by the results and/or references. Please, consider providing more information to justify this sentence or remove it from the paper.

Comment 5

Regarding Figure  5 to 7.

Please, define how is defined/calculated the "Normal Stress Amplitude".

Why did the authors select to show FF040, FF041 and FF033 and not other graphs?

Taking into account that the vertical and horizontal axis of these graphs are the sameà consider to put all results together in the same graph.

Comment 6

The results obtained in the paper are strongly influenced by the fitting approximation chosen on section 2 (FF approximation in this case). Nevertheless, the justification of the selection os FF approximation is not properly explained:

"To get the reasonably set fatigue strengths for each load case, another approximation than the Kohout-Věchet model should be applied. To increase the multiaxial fatigue strength analyses reported hereafter also to the lifetimes 1, 2, 5 and 10 million cycles, the FF approximation was chosen."

Comment 7

Consider to change the name of section 3. "Calculations" is too much general (maybe: "Multiaxial fatigue criteria").

Comment 8

"The validation program of multiaxial fatigue strength criteria on the presented experimental data concerned 24 various calculation methods. This paper presents 11 of them…" à What about the other 13?

Comment 9

Consider putting together the information of tables 5 and 6, including mean values and standard deviations in the same table.

Comment 10

Although there are too many combinations of loading scenarios and fatigue parameters, which make practically impossible to be able to represent all comparisons of fatigue failure predictions versus experimental results, it could be interesting to include some graphs (at least for the most interesting failure criteria) to illustrate how the \Delta FI has been obtained, and the results obtained in each case.

This can be done by different ways:

  • Graphs representing Nexp (s1)-vs-Npredic(Sigma_eq) for a failure criterion and for all loading cases (in different colors).
  • Graphs representing Nexp (s1)-vs-Npredic(Sigma_eq) for an specific loading case, and for the different failure criterions (in different colors).
  • Graphs representing S-N curves predicted (???,?(??)) versus observer experimentally (s1)

Comment 11

The conclusions started with the sentence: "The analysis presented at this paper focus on validating 11 different multiaxial fatigue strength criteria on the own test set composed of 34 S-N curves describing the fatigue response for different multiaxial load cases." But the 34 S-N curves are not shown in the paper (consider include these graphs in annexes).

"The experimental results described in the previous paper were extended by further 10 experimental test cases." à Avoid to include references in the conclusions.

Author Response

Comment 1

Title: Consider to modify the title of the paper. There is not a validation on the paper, only a comparison under different fatigue criteria.

"Validation of multiaxial fatigue strength criteria on  specimens from structural steel in the high-cycle fatigue region"

Answer 1: Wikipedia is not the right scientific tool for information, but it is handy to get the quick information: “…Validation. The assurance that a product, service, or system meets the needs of the customer and other identified stakeholders.” (from https://en.wikipedia.org/wiki/Verification_and_validation). Here, the individual computational routines are the system, customer or stakeholder is anybody trying to use them. The need is to get the best possible prediction quality (a limited scatter in fatigue index errors).

Comparison would mean analyzing different features of the criteria, etc. But we are mostly focused on discussing how well the prediction complies with the experimental outputs. According to us, this approach is much closer to validation.

Comment 2

First of all, the paper presents an introduction to the problem, which includes a brief motivation related to the necessity of performing various load combinations of push-pull and torsion tests to measure the quality of a multiaxial fatigue criterion, and the credibility of the results available in the literature (lines 29 to 52). After that, the authors start to describe the motivation, objectives, results and conclusions of reference [13], which forces the reader to read that reference in detail in order to understand the scope, motivation and objectives of the present paper. Although it is interesting to relate the present paper with the results obtained on reference [13], the actual paper should not be presented as a "simple extension" of the previous paper, which seems to be the intention of the authors (lines 53 to 70).

The main objectives of the paper must be highlighted in the introduction, and the way to reach these objectives could be resumed on the introduction. Additionally, the authors could include a short paragraph describing the paper organization.

Answer: Thank you for noting this weakness. The introduction in the second half was reworked so that the motivation for the new paper and its content were better described.

Comment 3

The second section of the paper (2. Experiments) must be improved. Different information is mixed in the same section, and too many calls to reference [13] are included.

The reviewer suggests to divide the information into different subsections and enlarge the information provided (avoiding to force the reader to consult ref [13](when it is possible)) Example:

2.1 Material (lines 75 to 80)

2.2 Geometry (lines 72 to 75 and figure 1) à consider to include S1, S2 and S3 figures on the current paper.

2.3 Loading Scenarios (lines 83 to  83 and 101 to 113;  Figure 2 and Table 3) à Consider to homogenize the description of the load scenarios (description of new experimental load cases (lines 87 to 93) differs substantially to the description of previous load cases (lines 101 to 103). Try to explain the load cases better; as an example, there is no clear difference between the different load scenarios FF048; FF04; FF050 and FF051 (because all of them correspond to the same figure (see figure 2)).

2.4 Testing procedure (lines 106 to 120) à Please, try to divide the actual paragraph into different paragraphs to describe the different load conditions and machines used.

2.5 Obtention of local stresses (lines 133 to 143) à please include some graphics representing examples of the stress distributions obtained.

2.6 Fitting the S-N curves (lines 160 to 199) à  Describe the 3 alternatives considered to fit the S-N curves obtained

2.7 Discussión the results obtained on the experimental programme à Describe the results and discuss the main conclusions obtained in this section (including comparison with previous works (lines 144 to 157).

Answer: We have to thank you for your advice, though we at last decided differently in some cases. You are right that the section was not well organized and dividing it into multiple subsections helped definitively. We added new items to decrease the dependency on the previous paper – there are drawings of all specimens now. There is new Table 4, which describes the load levels on each acting channels. Additional information on strain gaging and its reference to FEA models is also attached.

In this moment, the paper is 20 pages long. In one research paper, we cannot substitute a complete technical report or some PhD thesis. We decided not to provide the information on the stress distributions – taking into account the multiple load cases tested, it would be really only an illustration without any real impact on comprehensibility. The discussion of the regressions is appended to the end of Sec. 2.3.

Comment 4

"The approximation quality of the three mentioned formulas can be compared in Figs. 5-7." à What is the meaning of "approximation quality"? Can be the quality of formulas measured in an objective way?

Answer: You are right that the quality is very general term. We added another Fig. 7, which compares R2 parameters, but the separate paragraph explaining its content notes the limitations of such comparisons.

"The functions of the first two formulas (Basquin – Eq. (1), Kohout-VÄ›chet – Eq. (2)) are shown also outside the interpolation domain of analyzed data, so that their trends were manifested better." à Are the trends better because the formulas allow the extrapolation? Consider rewriting the sentence because all types of extrapolations could not be considered "manifested better".

Answer: The formulation was misleading, it should not mean that these formulas are in any way better, the manifestation was expected to be “better”. Reworded.

"Due to the reason mentioned previously, the FF approximation is shown only until ?? = 107 cycles." à the reason of shown only to 10^7 is maybe related to Eq. 3, not to the previous phrase.

Answer: Eq. 3 has nothing in common with that formulation. But it was too clumsy, so it was reformulated.

"The application of other regression models (Stromeyerova [21], Palmgren [22], Gecks-Ochs [23]) was also evaluated but none of them provided better results as regards the approximation quality." à It is not demonstrated in the paper, nor justified by the results and/or references. Please, consider providing more information to justify this sentence or remove it from the paper.

Answer: You are right that it is not reasonable to state that if we do not provide any proof. This text part was removed.

Comment 5

Regarding Figure  5 to 7.

Please, define how is defined/calculated the "Normal Stress Amplitude".

Why did the authors select to show FF040, FF041 and FF033 and not other graphs?

Taking into account that the vertical and horizontal axis of these graphs are the sameà consider to put all results together in the same graph.

Answer: Well, the quality of the information provided by these three large figures was not good enough. We managed to get them into one figure with a better description. These load cases were selected to manifest some positive features of the FF method, and also because of their importance in the validation – whereas any potential error in multiaxial load cases affects only the evaluation of that load case, these mentioned load cases project into material parameters a, b, c and d of the multiaxial criteria. Obtaining an adequate regression for them is thus more important.

Comment 6

The results obtained in the paper are strongly influenced by the fitting approximation chosen on section 2 (FF approximation in this case). Nevertheless, the justification of the selection os FF approximation is not properly explained:

"To get the reasonably set fatigue strengths for each load case, another approximation than the Kohout-Věchet model should be applied. To increase the multiaxial fatigue strength analyses reported hereafter also to the lifetimes 1, 2, 5 and 10 million cycles, the FF approximation was chosen."

Answer: We added a separate paragraph to the end of Sec. 2.3, where we discuss the selection, which is not that straightforward if only Fig. 7 would be assessed. The observation that the FF approximation is providing reasonable output can be deduced also from the analysis of DFI errors at different lifetimes – this comment is also added into Sec. 4.

Comment 7

Consider to change the name of section 3. "Calculations" is too much general (maybe: "Multiaxial fatigue criteria").

Answer: Your proposal is really better, we accepted it.

Comment 8

"The validation program of multiaxial fatigue strength criteria on the presented experimental data concerned 24 various calculation methods. This paper presents 11 of them…" à What about the other 13?

Answer: We reworded that part, so that the reasoning (which has already been stated there partly) was more obvious. Well, many of those non-mentioned methods show inferior prediction quality. Some of those presented are not better, but they are often referred to, and it is thus reasonable to show their limitations.

Comment 9

Consider putting together the information of tables 5 and 6, including mean values and standard deviations in the same table.

Answer: We apologize, but we insist that there is no reason to mix those parameters together. Mean value and standard deviation present completely different statistical characteristics, so even if the tables would be joined in some way, the content would be identical, but optically it would be less comprehensive.

Comment 10

Although there are too many combinations of loading scenarios and fatigue parameters, which make practically impossible to be able to represent all comparisons of fatigue failure predictions versus experimental results, it could be interesting to include some graphs (at least for the most interesting failure criteria) to illustrate how the \Delta FI has been obtained, and the results obtained in each case.

This can be done by different ways:

  • Graphs representing Nexp (s1)-vs-Npredic(Sigma_eq) for a failure criterion and for all loading cases (in different colors).
  • Graphs representing Nexp (s1)-vs-Npredic(Sigma_eq) for an specific loading case, and for the different failure criterions (in different colors).
  • Graphs representing S-N curves predicted (???,?(??)) versus observer experimentally (s1)

Answer: Thanks for interesting proposals. The description of the computation process was largely modified so that it was clear that we are dealing with fatigue strengths at given lifetimes and no lifetime predictions are performed. This fact removes the first two choices you proposed.

The last one is however a very good idea, which could be nice to observe in graphs. On the other hand, it would demonstrate per one graph only the prediction quality for one load case (and only several prediction criteria could get in, otherwise it would not be readable). We assume that output like that would be nice for some PhD thesis, but it cannot form a part of a paper in journal, where the space is limited. In that way the tables allow us to save the space efficiently, while the trends for individual lifetimes and individual methods are still nicely observable.

Comment 11

The conclusions started with the sentence: "The analysis presented at this paper focus on validating 11 different multiaxial fatigue strength criteria on the own test set composed of 34 S-N curves describing the fatigue response for different multiaxial load cases." But the 34 S-N curves are not shown in the paper (consider include these graphs in annexes).

Answer: We noted that previously – the scope of testing is too great to get it reasonably packed into one journal paper. Graphs are inefficient to cope with that. We decided to add an additional table, which describes the parameters of FF regressions for each load case also together with stating the interpolation region. This kind of information is comparable with your request.

"The experimental results described in the previous paper were extended by further 10 experimental test cases." à Avoid to include references in the conclusions.

Answer: Thank you for your wise advice.

Reviewer 2 Report

The paper deals with characterising and estimating the fatigue strength of non-welded structural steel under several multi-axial load cases. After justification of a proper representation of the S-N curve, eleven different methods for assessing multi-axial fatigue are applied and compared, leading to suggested methods for use.

The paper is interesting because the research bases on a sound and comprehensive database of experiments. Especially, the load combination of external loading and pressure is investigated and reported rarely. The applied method appear reliable. Influencing effects and different approaches to consider mean stress influences are discussed. The drawn conclusions are traceable with respect to the presented results.

Please consider when reviewing the manuscript:

  • Section 2: Please state if the combined loads acts simultaneously and which ratio of corresponding stresses (for instance for FF001: sigma(F)/tau(MK)) was realized.
  • Fig. 5 to Fig.7: How is normal stress amplitude defined? To which direction is the stress normal? Does the normal stress corresponds to the stress tensor derived by FE analysis? Please clarify.

Author Response

…Please consider when reviewing the manuscript:

  • Section 2: Please state if the combined loads acts simultaneously and which ratio of corresponding stresses (for instance for FF001: sigma(F)/tau(MK)) was realized.
    Answer: Though this request seems simple to implement, it is not. The lack of the relation between individual load channels does not concern only axial-torsion superposition, and the way some of the local stresses are imposed further complicate any attempt for a simple description. This was the reason, why we were originally quite brief. At last, the request was solved by imposing the new Table 4, which describes the acting forces, moments and pressure on individual load channels. The reader can use them to compute local stresses, if desired.

  • 5 to Fig.7: How is normal stress amplitude defined? To which direction is the stress normal? Does the normal stress corresponds to the stress tensor derived by FE analysis? Please clarify.
    Answer: To cover the answer to the meaning of normal stress amplitude, several new paragraphs were introduced into Sec. 3 to describe the basic computational concepts these criteria follow.

If your question concerned only the vertical axis of Figs.5-7, this was a simple error. The graphs were reduced into a single graph and the captions were changed to refer to more reasonable variables.

Conclusion: Thank you for your precious comments, we hope the changes in the text invoked in our attempt to answer them make the paper more comprehensive

Reviewer 3 Report

Excellent study on different multiaxial loading cases and further analysis using different multiaxial failure criteria.

The manuscript shall be published 

Author Response

Excellent study on different multiaxial loading cases and further analysis using different multiaxial failure criteria.

The manuscript shall be published 

Answer: Thank you for your kind review.

Reviewer 4 Report

The manuscript presents the results of fatigue life prediction using selected high-cycle multiaxial fatigue models. The models were tested using experimental data coming from the authors' own research. Some part of the data was previously published in other papers. The experimental data regards the multiaxial fatigue tests conducted on ÄŒSN 41 1523 structural steel hollow specimens of various geometry, loaded with different combinations of axial loading, torsion, plane bending, and inner pressure.

The presented research is interesting and deals with an important engineering problem. However, clarity is missing in some parts of the manuscript. Thus, I recommend accepting the submitted work after major revisions. Below are detailed comments which authors should respond to.

  1. Please provide information if the ÄŒSN 41 1523 steel has a European equivalent grade.
  2. There are plenty of acronyms and abbreviations used. A list of them would ease reading and understanding.
  3. On page 2, symbols "D" and "d" are used for diameters of specimens. Please clearly indicate they are outer and inner diameters.
  4. Figure 2 would be more informative if there were types of specimens, i.e. S1, S2, S3, indicated. Also, specimen type could be introduced in the caption of Figure 3.
  5. How were the loadings applied, measured, and controlled for different specimens and fixture configurations? Were the FEM calculations verified experimentally, for example using strain gauges? More information should be provided on page 6.
  6. Figures from 5 to 7 present the results of various fatigue curves fitting to the experimental data. The authors further discuss the quality of fitting. However, quantitative information is missing. The values of the coefficient of determination R2 is necessary, and any further information is recommended.
  7. The term "repeated loading" is used to name some types of loading. Does it concern the loading changing from zero to maximum? If so, the term "pulsating loading" is recommended.

Author Response

The manuscript presents the results of fatigue life prediction using selected high-cycle multiaxial fatigue models. The models were tested using experimental data coming from the authors' own research. Some part of the data was previously published in other papers. The experimental data regards the multiaxial fatigue tests conducted on ÄŒSN 41 1523 structural steel hollow specimens of various geometry, loaded with different combinations of axial loading, torsion, plane bending, and inner pressure.

The presented research is interesting and deals with an important engineering problem. However, clarity is missing in some parts of the manuscript. Thus, I recommend accepting the submitted work after major revisions. Below are detailed comments which authors should respond to.

  1. Please provide information if the ÄŒSN 41 1523 steel has a European equivalent grade.

Answer: The equivalents were added to Sec. 2 and also to the conclusion.

  1. There are plenty of acronyms and abbreviations used. A list of them would ease reading and understanding.

Answer: The section Nomenclature was added to the paper.

  1. On page 2, symbols "D" and "d" are used for diameters of specimens. Please clearly indicate they are outer and inner diameters.

Answer: This information was added to the nomenclature and also to the text parts, where the diameters are discussed the most.

  1. Figure 2 would be more informative if there were types of specimens, i.e. S1, S2, S3, indicated. Also, specimen type could be introduced in the caption of Figure 3.

Answer: Both your comments were solved according to your request.

  1. How were the loadings applied, measured, and controlled for different specimens and fixture configurations? Were the FEM calculations verified experimentally, for example using strain gauges? More information should be provided on page 6.

Answer: Indeed, individual load cases were equipped by strain gages so that the results of FE-analyses could be verified. A new paragraph on that point was added to Sec. 2.2.

  1. Figures from 5 to 7 present the results of various fatigue curves fitting to the experimental data. The authors further discuss the quality of fitting. However, quantitative information is missing. The values of the coefficient of determination R2 is necessary, and any further information is recommended.

Answer: Such comparison is tempting, and we provided it in a new figure. It anyhow does not provide the complete information – the sets of data points used for each of the regression differ, as it is explained in the accompanying text.

  1. The term "repeated loading" is used to name some types of loading. Does it concern the loading changing from zero to maximum? If so, the term "pulsating loading" is recommended.

Answer: We disagree with your proposal, because pulsating loading refers to the cases, where the lower load is non-zero and of the same sign as the maximum load. But disputes could be led on that point, so we decided to clarify this item in the Nomenclature, section Indexes.

Conclusion: Thank you for your comments. We believe that thanks to responding to them in the text, we made the paper more comprehensive for readers.

Round 2

Reviewer 1 Report

The reviewer wants to acknowledge the author's effort to improve the paper in terms of quality, organization and explanation of the results.

The reviewer's opinion is that, although not all of his considerations have been implemented in the new version of the paper, the authors have answered all the reviewer's comments and justified their decisions related to that. For that reason, the reviewer suggests to Accept the paper in present form.

Reviewer 4 Report

The authors provided extensive response to all my comments, and a significant number of changes and improvements was introduced into the manuscript. In my opminiow the manuscript is ready for publication. Congratulations on the excellent work.